# Snacking Behaviours of Australian Young Adults: Secondary Analysis of the MYMeals Cross-Sectional Study

**DOI:** 10.3390/nu15204471

**Published:** 2023-10-22

**Authors:** Jae Youn (Lisa) Han, Katrina Morris, Lyndal Wellard-Cole, Alyse Davies, Anna Rangan, Margaret Allman-Farinelli

**Affiliations:** 1Discipline of Nutrition and Dietetics, Sydney Nursing School, Faculty of Medicine and Health, The University of Sydney, Sydney, NSW 2006, Australia; jhan2593@uni.sydney.edu.au (J.Y.H.); kmor4184@uni.sydney.edu.au (K.M.); lyndalw@nswcc.org.au (L.W.-C.); alyse.davies@sydney.edu.au (A.D.); anna.rangan@sydney.edu.au (A.R.); 2Cancer Prevention and Advocacy Division, Cancer Council NSW, Sydney, NSW 2011, Australia; 3Charles Perkins Centre, The University of Sydney, Sydney, NSW 2006, Australia

**Keywords:** snack, snacking behaviours, young adults, Australia, energy-dense nutrient-poor snack foods, discretionary foods, MyMeals

## Abstract

Snacking outside main meals may contribute to the high intakes of discretionary foods (i.e., unhealthful foods) among young adults. This study assessed the snacking behaviours of Australian young adults including the contribution of snacking to energy and nutrient intakes, the main foods consumed, and portion sizes. A secondary analysis of the MYMeals study of adults aged 18–30 years who consumed at least one snack food during the recording period (n = 889) was conducted. All food consumed over 3 consecutive days was recorded using a purpose-designed smartphone app. Snack foods contributed 13.2% of energy, 23.4% of total sugars, and 16.2% of saturated fat. Females consumed more energy (13.8% vs. 12.2%, *p* = 0.007) and total sugars (25.8% vs. 20.8%, *p* = 0.009), from snacking than males. Fruit (20.2%), chocolate (9.9%), cake-type desserts (8.4%), sweet biscuits (6.1%), and ice-cream-type desserts (5.6%) were the most frequently consumed snacks by young adults. The median portion sizes for the top five snack foods consumed were fruit—106 g (IQR: 73), chocolate—26 g (IQR: 36), cake—95 g (IQR: 88), sweet biscuits—26 g (IQR: 29), and ice cream—75 g (IQR: 42). The current findings may inform population-wide strategies to encourage healthful snacks such as fruit, inform portion control by individuals, and persuade the food industry to reduce the serving size of discretionary snack foods such as cake.

## 1. Introduction

Worldwide prevalence of an overweight status and obesity has nearly tripled since 1975, Ref. [1], and in Australia, the most recent health survey, published in 2018, showed two in three adults were living with an overweight status or obesity with a large increase among 18- to 24-year-olds (from 39 to 46% in 3 years) [2]. The aetiology of this obesity epidemic is complex with a multitude of contributing factors, including both non-modifiable risk factors, such as genetics, and modifiable risk factors, like eating and exercise behaviours [3,4]. Among these factors is the high consumption of energy-dense nutrient-poor foods, typically known as discretionary foods in Australia, enabled with an obesogenic food environment predominated with such foods. This altered food environment and other factors have seen younger adults gain weight more rapidly than their parents’ generation [5].

In recent years, there has been a growing interest in how snacking habits can influence individuals’ diet composition and weight gain. A snack is defined as an eating occasion between the meals of breakfast, lunch, and dinner and could be healthful (defined as from the five food groups in Australia) or unhealthful food (discretionary food, i.e., high in saturated fat, and/or added sugars and/or sodium) [6]. Young adults consistently report higher rates of meal skipping and increased snacking occasions than other age groups [7,8,9,10]. It is commonly regarded as a dietary behaviour that contributes to an overweight status and obesity, although the effects of snacking on diet quality, nutrient and energy intake, and weight status are unclear [9]. Studies in the US have shown unhealthy snacking in college students is a contributing factor to poor eating habits and weight gain [11,12]. In Brazil, a study looking at snacking in young adults found that the top three snacks were cookies, sugar-sweetened beverages, and sweets and other desserts [13]. In the 2011-12 Australian National Nutrition and Physical Activity Survey (NNPAS), snacks contributed to 50% of daily discretionary food intake for males and 45% for females aged 19 to 30 years [14]. 

Some studies report the inclusion of snacks has alleviated potential digestive and metabolic overload due to fewer heavier meals and contributed to meeting recommendations for food groups (e.g., fruits and vegetables), and to having a more nutrient-dense diet [15,16,17]. In Australian young adults aged 19 to 30 years in 2011–2012, snacks made a small contribution to reaching daily targets for vegetables (3.5% males, 2.7% females) but a larger contribution to fruit intake (39.5% males and 33.4% females) [14]. 

The last available national dietary intake assessment in Australia is more than 10 years old. A more recent survey, the MYMeals survey (2017–2018), was conducted among 18- to 30-year-olds in Australia’s most populous state, New South Wales (NSW). This cross-sectional survey collected dietary data from 1001 young adults who recorded all food and beverages consumed over 3 consecutive days [18]. The aim of the present study was to conduct a secondary analysis of the MYMeals survey data to understand snacking behaviours in terms of energy, protein, total sugars, sodium, and saturated fat contribution to the total diet; the most popular snack foods; portion sizes; and differences by gender, age, socioeconomic status (SES), and Body Mass Index (BMI) categories. 

## 2. Materials and Methods

This secondary analysis utilized data from a cross-sectional survey, the MYMeals study, for which the protocol was previously published [18]. It included collection of 3-day dietary intake data from 18- to 30-year-olds across NSW, Australia. The study was approved by the University of Sydney Human Research Ethics Committee (project 2016/546).

### 2.1. Participants

Participants who met the following criteria were included in the study: (1) aged 18–30 years; (2) English-speaking and -writing; (3) owner of a smartphone; and (4) consumed at least one beverage, meal, or snack outside of home per week. Participants who consumed at least one snack (excluding beverages only) during the 3 days were included in the secondary analysis for this study. See Figure 1 for a flow chart of the study design. The MYMeals study aimed to recruit equal numbers of males and females and an even distribution of participants in the 18-to-24-year and 25-to-30-year age groups and of participants across the socioeconomic and geographic (metropolitan and non-metropolitan) groups using stratified recruitment [18]. 

### 2.2. Recruitment

A range of strategies were used to recruit participants including free and paid social media, letters mailed using a list compiled by the Australian Electoral Office, university electronic notice boards and newsletters, and community fundraising events held by Cancer Council NSW. Participants completed a screening questionnaire and those meeting inclusion criteria proceeded to provide further demographic details and received instructions about downloading and using a validated purpose-designed app to record all food and beverages consumed over a prescribed 3-day period with weekdays and weekends spread across the population.

### 2.3. Assessment of Dietary Intake 

Data from the MYMeals study were collected using a purpose-designed smartphone app, Eat and Track (EaT), and its validation was previously published [19]. Participants recorded all foods and beverages consumed, including the amounts and the location where the food was prepared, into the EaT app for 3 specified, consecutive days. The app enabled them to record meal occasions, including breakfast, lunch, dinner, and all snack and beverage consumption using a designated entry button. The app supported searches for common foods and commercial brand names and had an extensive database of menu items from fast food outlets, which was developed by Cancer Council NSW and The George Institute for Global Health’s fast-food database [18]. Other foods were from the database of the Australian Bureau of Statistics’ AUSNUT 2011–2013 database with some customization of food names to enable easier identification [18]. If participants were unable to find an item in the database of the app, freeform text entry was permitted, and nutrient values were manually entered by research dietitians.

The portion size of each item consumed was recorded in grams (g) or millilitres (ml) or selected from a predefined list based on portion sizes commonly consumed by males and females aged 18 to 30 years in the most recent NNPAS. 

### 2.4. Demographic Classification

Participants also completed a demographic questionnaire on the final day of dietary recording. The postcode of the area where the participant resided was collected and used to define SES by the Index of Relative Socio-economic Advantage and Disadvantage (IRSAD), which is published by the Australian Bureau of Statistics and accounts for both advantage and disadvantage [18]. BMI was calculated using self-reported height and weight, which was shown to be a valid measure in this study [18].

The demographic variables used for this secondary analysis were (1) age with two categories of (i) 18–24 years old and (ii) 25–30 years; (2) gender with two categories of (i) male and (ii) female; (3) SES with two categories of (i) low indicating those dwelling in lower SES areas (deciles 1–5 IRSAD) and (ii) high indicating those dwelling in higher SES areas (deciles 6–10 IRSAD); and (4) BMI with four categories of (i) underweight indicated as those with BMI of less than 18.5, (ii) healthy weight with BMI between 18.5 and 24.99, (iii) overweight with BMI between 25 and 29.99, and (iv) obese with BMI more than 29.99 [18]. 

### 2.5. Data Cleaning

For this study, beverages were excluded from the analysis as they were reported in a previous publication [20]. Milk consumed with cereal as a snack was included but milk as a beverage was excluded. 

### 2.6. Food Categorisation

The methodology described in the Australian Health Survey: Users’ guide 2011–2013 was followed to categorise each snack food into food groups [21]. The specific categories included in this study were the five food groups—fruits, vegetables and legumes, grain (cereal) foods, meat and alternatives, and dairy and alternatives—and the discretionary foods. Each food item was categorised, using the published food code list at the minor group (five-digit code) level and the unique code (eight-digit) level for some mixed dishes [21]. 

### 2.7. Statistical Analysis

The average intake of energy (kJ) and nutrients including protein (g), total sugars (g), sodium (mg), and saturated fat (g) from the consumption of snack foods for each participant over the 3 days was calculated. The percentage (%) contribution of snacks (excluding beverages) to the total daily energy and nutrient intakes was determined. 

As energy and nutrient analyses can be affected by energy misreporters, these participants were identified using the ratio of energy intake (EI) to basal metabolic rate (BMR). Participants who consumed an average EI over the 3 days of less than 1.0 × BMR estimated using the Schofield equation were flagged as low-energy reporters, and if they reported more than 2.4 × BMR, they were considered high-energy reporters [22]. Thus, those with EI:BMR of 1.0 to ≤2.4 were acceptable energy reporters. The percentage (%) contribution of snacks to the total daily energy and nutrient intakes of acceptable reporters was also calculated. 

The median and interquartile range (IQR) of the average intake of energy (kJ) and nutrients including protein (g), total sugars (g), sodium (mg), and saturated fat (g) from the consumption of snacks for each participant over the 3 days were calculated for both total and acceptable energy reporters. 

Each snack food was labelled as either discretionary or five-food-group food using the discretionary food list that was used for food categorisation. Using this, the proportion (%) of discretionary versus five-food-group food from total snacks consumed was calculated.

The top ten snack foods were determined with frequency of consumption and percentage energy contribution to total snack foods. Frequency was determined by counting the number of foods consumed as snacks in the specific food group category to the total number of snack foods consumed among all snack consumers. For energy contribution (%), the sum of energy (kJ) intake from each food group category to the total energy (kJ) from all snack foods consumed was calculated. Based on these, the top ten snack foods were ranked and recorded. 

Portion sizes in grams were determined for the top five most frequently consumed discretionary foods for snacking because these foods are most likely to have deleterious effects on diet quality. Additionally, the average kJ consumed for each snack food group was calculated, and the number of servings was identified using the serving sizes described in the Australian Guide to Healthy Eating (AGHE) with 1 serving = 600 kJ [6]. 

For assessing any differences in contribution of snack foods to daily total energy and nutrient intakes and portion sizes by age, gender, and SES, the Mann–Whitney U test was used and the Kruskal–Wallis test was used for BMI. For investigating differences in proportions of discretionary foods versus five-food-group foods consumed for snacking by age, gender, SES, and BMI, the Chi-squared test was used. Total daily energy, protein, total sugars, saturated fat, and sodium intakes were compared with the quartile of energy consumed from snacks using an analysis of variance and *p* for the trend. All statistical analyses were conducted using IBM SPSS Statistics for Windows (2017), Version 25.0. (Armonk, NY, USA: IBM Corp and Microsoft Excel (2018), Version 2302).

## 3. Results

### 3.1. Characteristics of Snack Consumers

Among the 1001 participants of the study, 889 (89%) consumed snack foods at least once during the 3-day recording period and their characteristics are shown in Table 1. The sample comprised 58% females and 54% of participants were aged 18–24 years. Most participants had a healthy BMI (56%), with 38% classified as overweight or obese. In this sample, 41% were classified as low SES. Appendix A compares characteristics of the acceptable energy reporters.

### 3.2. Contribution of Snacking to Total Energy and Nutrient Intakes

Table 2 shows the median contribution (%) of snacking to total energy and nutrient intakes including protein, total sugars, sodium, and saturated fat among participants who consumed at least one snack during the 3-day recording period (n = 889), stratified by participant characteristics. Overall, snacking contributed to 13.2% of total energy intake, 6.2% of protein, 23.4% of total sugars, 7.3% of sodium, and 16.2% of saturated fat intake. Females had a significantly higher contribution to energy (13.8% versus 12.2%, *p* = 0.007) and total sugars (25.8% versus 20.7%, *p* = 0.009) from snack foods compared to males. The contribution of energy and nutrients from snacking to total energy and nutrient intake was similar among age, SES, and BMI groups.

When only acceptable reporters were included (n = 589), snacking appeared to make a higher contribution to overall energy and nutrients. Females had a significantly higher contribution to energy (*p* = 0.005), total sugars (*p* = 0.002), and saturated fat (*p* = 0.016) from snacking than males (see Appendix A).

### 3.3. Average Total Energy and Nutrient Intake from Snacking 

Table 3 shows the median intakes of total energy (kJ) and nutrients from snacking among all participants consuming snacks stratified by participant characteristics. Overall, participants consumed 993 kJ, 4.8 g of protein, 15.1 g of total sugars, 164 mg of sodium, and 4.1 g of saturated fat from snacks per day. None of the intakes demonstrated statistically significant differences by demographic factors. 

When only acceptable energy reporters were included, slightly higher intakes of energy, protein, sugars, sodium, and saturated fat from snacking were observed than for the total sample (see Appendix A). 

### 3.4. The Proportion of Discretionary Food Consumed as Snacks 

Table 4 presents the number of snacks identified as discretionary foods versus five food groups and the percentage to the total number of snacks (n = 4685) by age, gender, SES, and BMI. A higher proportion of snacks was identified as discretionary among participants residing in lower SES areas compared to those living in higher SES areas (61 versus 54%, *p* < 0.001), and by those with BMI classified in the obese category (65%) compared with those in the underweight, healthy weight, or overweight categories (*p* < 0.001).

### 3.5. Top Ten Food Sources by Frequency and Energy Contribution

The top ten snack food types by frequency of consumption (%) and contribution to total energy intake among all 889 participants are shown in Table 5. The five most frequently consumed snack foods were fruit (20.0%); chocolate (9.9%); cakes, doughnuts, muffins, pastries, slices, and sugar desserts (8.4%); sweet biscuits (6.1%); and ice cream, cream, soft serve, and frozen yoghurt (5.6%). The snack foods contributing the most to total energy intake were cakes, doughnuts, muffins, pastries, slices, and sugar desserts (17.6%); chocolate (11.1%); fruit (7.8%); crisps and extruded snacks (7.2%); and ice cream, cream, soft serve, and frozen yoghurt (6.7%).

### 3.6. Portion Size (g) of Top Five Most Common Discretionary Foods Consumed as Snacks

Table 6 presents the median portion sizes (g) of the top five most consumed snacks. These included (1) cakes, doughnuts, muffins, pastries, slices, and sugar desserts; (2) chocolate; (3) fruit; (4) crisps and extruded snacks; and (5) ice cream and frozen yoghurt. Participants with BMIs in the obese category had significantly larger median portion sizes of cakes, doughnuts, muffins, pastries, slices, and sugar desserts (117 g, *p* = 0.021), and chocolate (41.5 g, *p* < 0.01), compared to those with underweight, healthy weight, or overweight BMIs. Males had significantly larger median portion sizes of crisps (45 g, *p* = 0.048), ice cream, and frozen yoghurts (89 g, *p* = 0.002) than females.

### 3.7. Portion Sizes of Top Five Most Common Discretionary Foods Consumed as Snacks

Figure 2a illustrates the portion sizes of chocolate by BMI categories. This figure shows that of those who consumed chocolate as snacks at least once during the 3-day recording period, the largest proportion of participants in the obese and overweight categories (61% and 55%) consumed one or more AGHE servings per day while the largest proportion in the underweight and healthy weight categories consumed less than one serving per day. Figure 2b shows the serving sizes of cakes, doughnuts, muffins, pastries, slices, sugar desserts, and puddings consumed by BMI categories. This figure shows that of those who snacked on cakes, doughnuts, muffins, pastries, slices, sugar desserts, and puddings at least once during the 3-day recording period, 68% of the participants classified in the obese category were having more than two AGHE servings followed by 59% in overweight, 51% in healthy weight, and 33% in underweight categories. 

Figure 3a illustrates the portion sizes of ice cream, cream, soft serves, and frozen yoghurt by gender. This figure shows that of those who consumed ice cream, cream, soft serve, and frozen yoghurt as snacks at least once during the 3-day recording period, the largest proportion of those in the male group (45%) and female group (41%) were having between one to two AGHE servings per eating occasion. Figure 3b demonstrates the portion sizes of crisps and extruded snacks by gender. This figure shows that of those who consumed crisps and extruded snacks as snacks at least once during the 3-day recording period, the largest proportion of males (46%) and females (39%) were having between one to two servings per day.

Table 7 shows the comparison of intakes of energy and nutrients by the quartile of energy participants consumed from snacks. Those who consumed the greatest percentage of energy from snacks (28%) also had the highest intakes of total energy and other nutrients with a positive trend of increasing consumption of all nutrients across quartiles when more energy was consumed from snacks (*p* < 0.001).

## 4. Discussion

The findings indicate that young adults, aged 18–30 years, are choosing some snack foods that are disproportionally contributing to saturated fat and total sugar intakes. Females consumed a greater percentage of their energy intake and sugars from snack foods compared with males. Fruit, a healthful food, was the most popular of all snacks. However, 7 of the 10 top snack food types contributing to energy intake were unhealthful discretionary food items. Chocolate, cakes, muffins and pastries, sweet biscuits, ice cream, and potato crisps were among the most popular and energy-contributing snack foods. Participants consumed portions of cakes and potato crisps exceeding one 600 kJ serving and those participants classified in the obese category consumed more of their snacks from the unhealthful foods and ate larger portions of cake and chocolate. Clearly there is an opportunity to improve the nutritional quality of snacking by encouraging more fruit and five-food-group foods such as vegetables, nuts, and seeds and less discretionary foods, and to examine appropriate portion sizes. 

Young adults have been identified as highly vulnerable to snacking, with growing independence, exposure to new social groups, and starting university or entering the workforce for the first time. These changes can be stressful and hinder consumption of healthful food. Interventions have demonstrated that adults can improve their fruit and vegetable intakes [23]. In addition to fruit, participants also snacked on nuts, seeds, and vegetables, which should be encouraged. Decreasing consumption of discretionary foods low in protein and fibre and replacing these foods with those from the healthful five food groups could contribute to better diet quality, and the satiation properties of protein and fibre are well documented to assist with appetite control [24].

Portion sizes of two of the top five discretionary snack foods varied significantly according to the BMI categories. There has been an increase in portion sizes between the last two national dietary surveys in Australia and these changes typically involve discretionary foods [25]. The portion size of cake increased the most by 950 kJ since 1995, with other categories such as ice cream and pizza significantly increasing as well. In contrast, typical portion sizes of pastries, snack food, and potato fries decreased by 10–40% over time, and no changes were found for biscuits and chocolate [25]. In the current study, cakes were a major contributor to energy as were chocolate and ice cream.

A systematic review on the effect of reducing portion sizes found that a slight reduction was unnoticeable to consumers and was associated with less weight gain compared to when larger portion sizes of food were consumed [26]. A review of RCTs (22 included studies) found that consuming larger portion sizes and greater eating frequencies were associated with higher energy consumption [27]. The relationship between snacking patterns and weight status is unclear throughout the literature and is likely dependent on the type of snacks consumed. Some studies report health benefits such as improved appetite control and body weight regulation, associated with snacking on whole foods high in protein, fibre, whole grains such as nuts, and yoghurt [28].

There were significant differences between low- and high-SES participants with those in low-SES areas consuming a higher proportion of snacks as discretionary foods. This is consistent with previous studies showing that low-SES groups had higher intakes of discretionary foods than those in higher SES, and discretionary foods were one of the main contributors to total energy intake [29]. Differences between genders and snacking behaviours have also been reported previously. Adolescent females had more frequent consumption of snacks than males and were more likely to skip breakfast and lunch, having less energy and nutrient intakes from main meals [30]. Similarly, in the current study, females consumed a greater percentage of their energy intake and sugars from snack foods compared with males. 

Health promotion programs should aim to reach young adults before these snacking behaviours become engrained habits, to prevent and reduce the occurrence of negative health outcomes later in life. Given the differences by sociodemographic characteristics, targeted programs should be stratified by gender, SES, and BMI classifications. As well as individual behaviour change, the food environment is an important determinant of food choices. It provides opportunities for menu improvements, clearer food labelling, price incentives, and the promotion of healthful foods instead of those less healthful [31]. In a small subsample of the MYMeals study, 133 participants wore cameras that captured images every 30 s, thereby providing objective evidence on consumption contexts [32]. It confirmed snacks had the highest energy density among eating occasions and when snacks were prepared outside the home, they were more likely to be discretionary [32].

The strength of this study includes sampling young adults from across NSW with attention to age, gender, and SES distribution. Also, as the data were stratified by demographic characteristics, this study was able to investigate the differences in snacking behaviours by demographic characteristics. There are some limitations of the current study. All data are self-reported and subject to bias. Cross-sectional data preclude causal conclusions to be drawn regarding snack consumption practices and BMI category. The prevalence of unacceptable energy reporters (34%) among snack consumers included in our study is of concern but the nutrient densities including the percentage contribution of snacking to total daily energy and nutrient intake are less affected by under-reporting [33]. Snack occasions were subjectively defined by each participant when recording their dietary data in the app so misclassification of meals may have occurred.

## 5. Conclusions

This present study provides insight into snacking patterns among young adults in Australia. Fruit was the most frequent snack food selected and could be further encouraged together with choices such as vegetables, nuts, and seeds. Overall, young adults consumed a small percentage, 13%, of their energy from snacks, but their food choices provided them with a greater proportion of saturated fat and total sugars that may impact negatively on diet quality. It should be noted that this is exclusive of beverages that may further add to energy intakes. Public health efforts to develop targeted recommendations on healthful snacking patterns and regulation of portion sizes of discretionary snack foods may enable better snacking behaviours among young adults.

## Figures and Tables

**Figure 1 nutrients-15-04471-f001:**
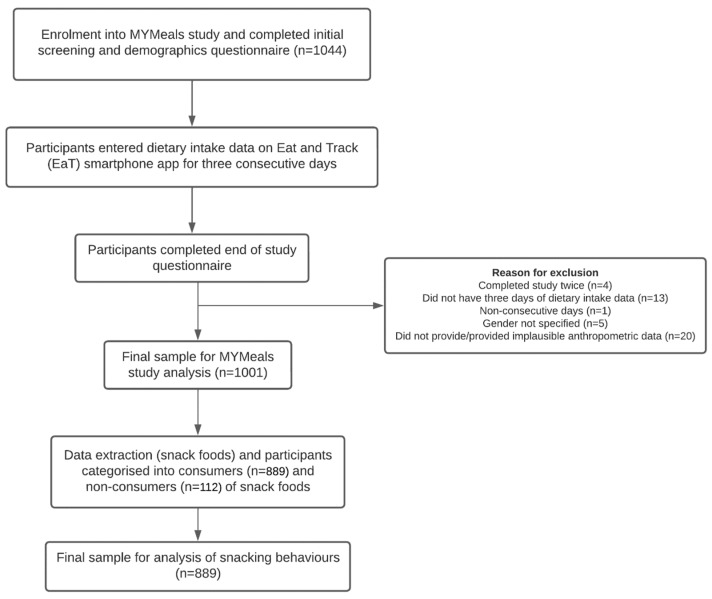
Flow diagram of the study design.

**Figure 2 nutrients-15-04471-f002:**
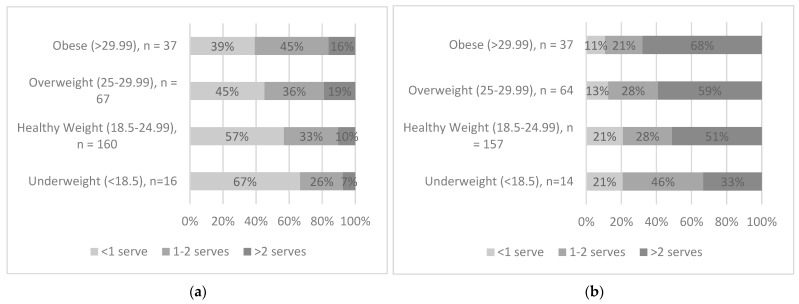
(**a**) Portion sizes of chocolate by BMI categories among those consuming. (**b**) Portion sizes of cakes, doughnuts, muffins, pastries, slices, sugar desserts, and puddings by BMI categories among those consuming. The number of servings was identified using the serving sizes described in the Australian Guide to Healthy Eating (AGHE) with 1 serving = 600 kJ [6].

**Figure 3 nutrients-15-04471-f003:**
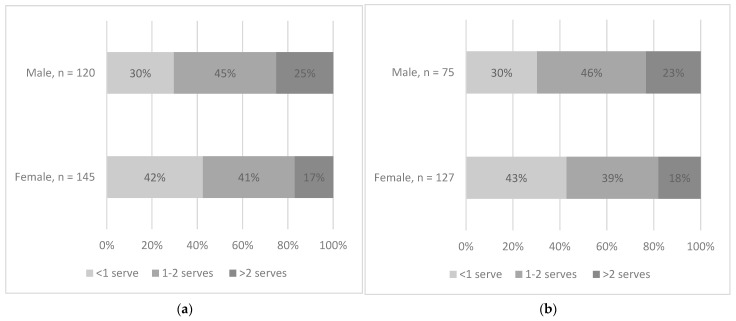
(**a**) Portion size of ice cream, cream, soft serves, and frozen yoghurt by gender among those consuming. (**b**) Portion size of crisps and extruded snacks by gender among those consuming. The number of servings was identified using the serving sizes described in the Australian Guide to Healthy Eating (AGHE) with 1 serving = 600 kJ [6].

**Table 1 nutrients-15-04471-t001:** Snack consumer demographic characteristics for consumers of snacks for the total sample and for those classified as acceptable energy reporters.

Participant Characteristics		Total Sample n = 889 (%)
Total sample		889
Gender	Female	520 (58)
	Male	369 (42)
Age group (years)	18–24	479 (54)
	25–30	410 (46)
Body Mass Index (kg/m^2^)	Underweight (<18.5)	47 (5)
	Healthy weight (18.5–24.99)	501 (56)
	Overweight (25–29.99)	224 (25)
	Obese (>29.99)	117 (13)
Socioeconomic status ^a^	Low	363 (41)
	High	526 (59)

^a^ From Socio-Economic Indexes for Areas Index for Relative Advantage and Disadvantage [18] based on residential postcode; lowest five deciles = low, highest five deciles = high.

**Table 2 nutrients-15-04471-t002:** Median (IQR) contribution (%) of energy, protein, total sugars, sodium, and saturated fat from snacking to total energy and nutrient intake in all participants (n = 889) who consumed at least one snack during the 3 days.

	Sample Size, n	Energy		Protein		Total Sugars		Sodium		Saturated Fat	
		Median (%)	IQR	Median (%)	IQR	Median (%)	IQR	Median (%)	IQR	Median (%)	IQR
Consumers	889	13.2	13	6.2	8.4	23.4	25.1	7.3	11.7	16.2	20.2
Age (years)											
18–24	479	13.5	12.6	6.2	8.2	23.4	25.6	7.4	11.1	16.6	20.7
25–30	410	12.9	13.4	6.4	8.9	23.2	24.9	7.3	12.5	15.7	19.8
*p*-Value		0.852		0.674		0.492		0.913		0.828	
Gender											
Male	369	12.2	12.8	5.7	9	20.7	21.6	6.8	12	15.3	19
Female	520	13.8	13.3	6.5	8	25.8	26.3	7.6	11.4	17	22.1
*p*-Value		0.007		0.082		0.009		0.167		0.111	
SES ^a^											
Low	363	13.5	14.4	6.5	8.8	22.9	25.9	7.9	12.8	16.5	20.6
High	526	12.9	12.4	5.9	8.2	23.9	24.5	7	11.1	15.8	19.7
*p*-Value		0.344		0.359		0.389		0.144		0.531	
BMI											
Underweight (<18.5)	47	12.8	11.9	5.7	7.7	24.8	23.5	6.8	10.4	13.5	14.2
Healthy weight (18.5–24.99)	501	13.6	13.3	6.4	8.7	23.8	25.1	7.6	11.7	16.6	22.2
Overweight (25–29.99)	224	12.8	12.2	6.5	8.9	22.9	23.8	7.3	12.4	16.6	17.7
Obese (>29.99)	117	12.5	13.7	5.4	7.8	22.2	26.4	6.1	11.9	14.7	17
*p*-Value		0.619		0.254		0.795		0.468		0.463	

^a^ SES is from Socio-Economic Indexes for Areas Index for Relative Advantage and Disadvantage [18] based on residential postcode; lowest five deciles = low, highest five deciles = high.

**Table 3 nutrients-15-04471-t003:** Median (IQR) of average intake of total energy, protein, total sugars, sodium, and saturated fat across 3 days from snacking by all participants (n = 889) who consumed at least one snack during the 3-day recording period.

Characteristics	Sample Size, n	Total Energy (kJ)		Total Protein (g)		Total Sugars (g)		Total Sodium (mg)		Total Saturated Fat (g)	
		Median	IQR	Median	IQR	Median	IQR	Median	IQR	Median	IQR
Consumers (total)	889	993	1192	4.8	7.5	15.1	20.3	164	295	4.1	6.1
Age (years)											
18–24	479	994	1173	4.7	7.0	15.7	19.9	164	293	4.2	6.2
25–30	410	985	1206	5.0	8.1	14.2	20.0	164	309	4.0	6.2
*p*-Value		0.97		0.24		0.33		0.97		0.66	
Gender											
Male	369	1043	1327	5.0	8.7	15.2	20.2	176	380	4.5	6.7
Female	520	975	1100	4.7	6.7	14.8	20.4	156	259	3.8	5.9
*p*-Value		0.40		0.48		0.87		0.48		0.07	
SES ^a^											
Low	363	1033	1259	5.0	7.0	15.5	20.5	177	313	4.1	6.8
High	526	983	1151	4.7	7.9	14.8	20.0	156	275	4.0	5.9
*p*-Value		0.48		0.26		0.66		0.16		0.98	
BMI											
Underweight	47	922	946	3.8	5.8	14.3	21.0	123	199	3.6	4.8
Healthy weight	501	1023	1168	4.8	7.7	15.2	19.9	166	281	4.2	6.1
Overweight	224	1030	1203	5.3	7.5	15.0	20.0	183	337	4.2	6.9
Obese	117	925	1333	3.9	7.2	14.3	23.6	117	310	3.7	6.8
*p*-Value		0.69		0.19		0.96		0.64		0.82	

^a^ Socio-Economic Indexes for Areas Index for Relative Advantage and Disadvantage [18] based on residential postcode; lowest five deciles = low, highest five deciles = high.

**Table 4 nutrients-15-04471-t004:** Counts and proportion of snack foods consumed that are discretionary vs. non-discretionary by age, gender, socioeconomic status, and BMI group.

Characteristics	Total Snacks, n	Discretionary, n (%)	Five Food Groups, n (%)	*p*-Value
Age				
18–24 yrs., n = 479	2448	1415 (58)	1033 (42)	0.253
25–30 yrs., n = 410	2237	1256 (56)	981 (44)	
Gender				
Male, n = 369	1830	1058 (58)	772 (42)	0.374
Female, n = 520	2855	1613 (56)	1242 (44)	
SES				
Low, n = 363	1790	1097 (61)	693 (39)	<0.001
High, n = 526	2895	1574 (54)	1321 (46)	
BMI				
Underweight, n = 47	260	138 (53)	122 (47)	<0.001
Healthy weight, n = 501	2813	1523 (54)	1290 (46)	
Overweight, n = 224	1108	664 (60)	444 (40)	
Obese, n = 117	504	346 (69)	158 (31)	

**Table 5 nutrients-15-04471-t005:** Top 10 snack foods consumed by frequency of consumption and % energy contribution to total snack foods.

	Snack Food	Frequency of Consumption (%)	Rank	Snack Food	Energy Contribution (%)
1	Fruit (fresh and dried)	20.2%	1	Cakes, doughnuts, muffins, pastries, slices, sugar desserts	17.6%
2	Chocolate	9.9%	2	Chocolate	11.1%
3	Cakes, doughnuts, muffins, pastries, slices, sugar desserts	8.4%	3	Fruit (fresh and dried)	7.8%
4	Sweet biscuits	6.1%	4	Crisps, extruded snacks	7.2%
5	Ice cream, cream, soft serve, frozen yoghurt	5.6%	5	Ice cream, cream, soft serve, frozen yoghurt	6.7%
6	Crisps, extruded snacks	5.6%	6	Sweet biscuits	5.9%
7	Nuts and seeds	4.6%	7	Nuts and seeds	5.3%
8	Vegetables	3.6%	8	Muesli bars, nut- and seed-based confectionary	2.7%
9	Lollies	3.1%	9	Potato products	2.5%
10	Muesli bars, nut- and seed-based confectionary	3.0%	10	Sweet breads	2.4%

**Table 6 nutrients-15-04471-t006:** Median (IQR) portion size (g) of top five food categories of snacks by total consumers, gender, and BMI.

		Food Category														
		Cakes, Muffins, Sweet Pastries, Sugar-Type Desserts			Chocolate			Fruit (Fresh and Dried)			Crisps, Extruded Snacks			Ice Cream and Frozen Yoghurt		
		n	Median (g)	IQR	n	Median (g)	IQR	n	Median (g)	IQR	n	Median (g)	IQR	n	Median (g)	IQR
Consumers ^a^		272	95	88	280	26	36	437	106	73	202	45	28	265	75	42
Gender	Female	170	95	92	164	26	34	275	95	72	127	40	30	145	74	41
	Male	102	95	82	116	26	36	162	117	69	75	45	25	120	89	59
	*p*-Value	0.794			0.621			0.057			0.048			0.002		
BMI	Obese	37	117	105	37	42	33	44	99	103	29	47	29	37	90	74
	Overweight	64	101	88	67	30	35	98	111	77	53	40	29	71	75	41
	Healthy weight	157	90	89	160	25	27	269	104	70	110	45	29	144	74	42
	Underweight	14	76	70	16	25	24	26	109	77	10	45	25	13	75	66
	*p*-Value	0.021			<0.01			0.656			0.899			0.090		

^a^ Consumers are the number of people who consume the snack and not the frequency of the consumption.

**Table 7 nutrients-15-04471-t007:** Associations between quartiles of energy intake from snacks for total energy and nutrient intakes. Values are means (standard error).

	Quartile 1	Quartile 2	Quartile 3	Quartile 4	*p*-Trend
	<524 kJ	524–992 kJ	993–1709 kJ	>1709 kJ	
n	223	222	222	222	
Total energy (%)	4.9 (0.2)	11.2 (0.3)	16.7 (0.3)	28.3 (0.6)	<0.001
Total energy (kJ)	6714 (188)	7651 (179)	8420 (149)	10243 (221)	<0.001
Protein (g)	80.3 (2.6)	84.6 (2.2)	89.9 (2.2)	103.2 (3.4)	<0.001
Total sugars (g)	51.2 (2.0)	68.4 (2.4)	76.0 (2.3)	105.5 (3.2)	<0.001
Saturated fat (g)	21.6 (0.9)	25.2 (0.9)	29.0 (0.7)	38.6 (1.2)	<0.001
Sodium (mg)	2244 (67)	2336 (72)	2682 (72)	3014 (88)	<0.001

Age (*p* = 0.779), gender (*p* = 0.22), socioeconomic status (*p* = 0.62), and BMI (0.710) were not significant across the quartiles.

## Data Availability

Data sharing is not available for this study due to ethics.

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
