# Peer review of "Snacking Behaviours of Australian Young Adults: Secondary Analysis of the MYMeals Cross-Sectional Study"

_nutrients, 2023, doi:10.3390/nu15204471_

Round 1
Reviewer 1 Report
I think this is an interesting paper, with a high sense of opportunity.
Nevertheless, the abstract, and the conclusions should be reworded/improved.
Abstract: should be clear about the purpose of the study. Lines 18-24 are results, they should not be in the abstract. There should be only the main conclusions.
Results: section 3.1 is not results, they should move on to the previous section.
Conclusions: should be deepened and demonstrate the relevance of the research.
Author Response
see attached. Thank you for your contribution.

Reviewer 2 Report
Snacking behaviour has become of greater interest as obesity and intake of processed foods have increased. The authors provide a new look into this subpopulation and snacking. A distinction by the authors between snacking foods and discretionary foods is confusing and unnecessary. If a food is eaten as a snack, then it is a snack. Not all snacks are unhealthy. The unhealthy nature of Australian snacks is a bit overstated due to this odd distinction of their sources. This needs to be corrected.
Abstract
Apparently the most highly consumed snacks were fruits. This contradicts the initial assertion that discretionary foods are all unhealthy and snacks are the leading contributors in this group. The authors should modify this Abstract to reflect this. One addition that would help would be to include the median portion size of fruit intake to compare with the other snacks reported. Also, the conclusions may need altering to reflect that not all snacks are unhealthy and consumption of certain discretionary foods may not need to be reduced but actually encouraged.
Line 18: The inclusion of beverages here implies that beverages were part of the snacks. It is import for readers to understand that this study does not include beverages.
Line 23: In line 19, snack foods are compared. In line 23, discretionary foods are compared. This is a confusing and unnecessary distinction. Please compare snack foods for both analyses.
Keywords
Since the study is a secondary analysis of the MYMeals study, it might be appropriate to include “MYMeals” in the keywords.
Introduction
It would be helpful to understand the distinction between a snack food and a discretionary food. What is a discretionary food that is not a snack?
Line 64: It is interesting that the authors suggest that achieving about 1/3 of the target fruit intake through snacking is a “still small contribution.” This is 1/3 of the target, apart from 3 regular meals and beverages. I would call that some pretty healthy snacking. By what standard is this small? Possibly a change of language here?
Results
Table 6 is titled, “Median (IQR) portion size (g) of top five discretionary food categories of snacks…” From left to right the table includes the top foods from the right side of Table 5 with one glaring exception-fruit. Again, by switching from “snack foods” in Table 5 to “discretionary foods” in Table 6, the authors were able to omit fruit from Table 6. Please correct this by changing the title of Table 6 and including the top five snack foods from Table 5 into Table 6.
Figures 1 and 2 are a bit misleading. To the casual reader, it is assumed that the intake group “<1” includes zero intake. But this is not true. The population of these bars are a subset of a subset. Among all 18–30-year-olds, the study included only those that snacked, not counting beverages. Each bar of these figures only includes the subset of snackers that ate that specific snack at least once in 3 days. At a minimum this can be clarified by amending the caption to read something like, “Portion sizes of chocolate by BMI categories among chocolate snackers.” Another solution would be to label the lowest group “1/3-1”.
Discussion
Lines 308-310: One of the confusing and rather unnecessary distinctions raised in this paper is between snack foods and discretionary foods. This second sentence of the Discussion is an example. “This was unsurprising considering seven of the 10 top snack food types contributing to energy intake were discretionary food items.” As if this distinction between snack and discretionary foods is the second most important finding of the study. The authors have not made such a pressing distinction between these two that I as a reader am impressed by this finding. As a reader, I am interested in snacking. The authors need to make a much stronger case for non-snack discretionary foods to hold my interest.
The finding that 13.2% of energy intake in this group is from snacks is labeled “excessive consumption of discretionary snack foods”. As a reader, a nutritionist, and an observer of the common obesity problem, this value does not strike me as either excessive or alarming. This is roughly a third to a half of what is consumed at a meal. And this is only the population that consumes at least one snack every three days. What may be more interesting is the high consumers, those in the top quarter of consumers. The findings of Figure 1 are more interesting in that regard. Could quartiles of consumers be compared, for example?
References
A number of small inconsistencies in references need to be addressed. Journal names are inconsistently abbreviated. Periods and italics are used inconsistently. In author lists, commas and semi-colons are used inconsistently.
Author Response
Thank you very much and see the attached file.

Round 2
Reviewer 2 Report
Clarifications and corrections have been made.